# Dynamics of factors associated with neonatal death in Madagascar: A comparative analysis of the 2003, 2008, 2021 DHS

Sedera Radoniaina Rakotondrasoa[1,2]*, Kadari Cissé[3], Tieba Millogo[4], Hajalalaina Rabarisoa[1,2], Felix Alain[1], Seni Kouanda[3,4], Julio El-C. Rakotonirina[1,2]

**1** Faculty of Medicine of Antananarivo, University of Antananarivo, Antananarivo, Madagascar, **2** Teaching Hospital of Care and Public Health, Analakely (CHUSSPA), Antananarivo, Madagascar, **3** Institute of Research in Health Sciences (IRSS), National Center for Scientific and Technological Research, Ouagadougou, Burkina Faso, **4** African Institute of Public Health (IASP), Ouagadougou, Burkina Faso

* sederarado.srr@mail.com

## Abstract

Neonatal mortality remains a major public health challenge, as reductions have stagnated worldwide despite cost-effective interventions in recent years. The temporal evolution of its determinants is insufficiently studied. This study aimed to analyze the dynamics of factors associated with neonatal death in Madagascar between 2003 and 2021. A secondary analysis was conducted using data from the 2003, 2008, and 2021 Demographic and Health Surveys (DHS) of Madagascar. The study population is focused on children under the age of 5 years at the time of these surveys. The death of a newborn within 30 days after birth constitutes the outcome variable. A multilevel binomial logistic regression was performed. The number of children under 5 included in the analysis were 5,415 in 2003, 12,448 in 2008 and 12,399 in 2021. The prevalence of neonatal deaths was 3.1% in 2003, 2.4% in 2008, and 2.6% in 2021. Persistent significant associations with neonatal death were observed for low birth weight, lack of breastfeeding, medium-sized households, large households, and high birth weight. A loss of statistical significance of the association with neonatal death over time was observed for a birth interval of 2–3 years and 4 years and more, mother's age 40–49 years, and use of mosquito net by the mother. In 2021, new significant associations with neonatal mortality were identified in the province of Toliara, absence of geographic barriers to healthcare access, 4–7 ANC visits, and supervised delivery. The factors associated with neonatal mortality, which have worsened over time in Madagascar, include birth weight abnormalities and delivery in the presence of qualified personnel. This deterioration underscores the urgency of improving the quality of perinatal care in healthcare facilities, beyond mere geographical accessibility.

**Data availability statement:** The data supporting the findings of this study were obtained from The Demographic and Health Surveys (DHS) Program (www.dhsprogram.com). Specifically, the analysis utilized the Madagascar Birth Recode (BR) files contained within the archives MDBR42DT.ZIP (2003-04 survey), MDBR51DT.ZIP (2008-09 survey), and MDBR81DT.ZIP (2021 survey). These microdatasets are not publicly shared or redistributed due to participant confidentiality and the DHS Program's terms of use. However, they are available for legitimate research purposes through the DHS Program's established access procedures. Researchers wishing to access these datasets are required to register individually on the DHS Program website and follow a specific process. First, they should visit the DHS Program's dataset portal at https://dhsprogram.com/data/available-datasets.cfm , select Madagascar as the country of interest, and locate the surveys conducted in 2003-04, 2008-09, and 2021. Within each survey page, researchers need to click on "Data Available," register or log in to their account, and create a new research project request. In the request, they must clearly state the purpose of verification (referencing this article is recommended), specify the required Madagascar BR datasets (MDBR42, MDBR51, MDBR81), and agree to the DHS terms of use. Once approved by The DHS Program, the dataset archives will be made available for download through the researcher's authorized account. The corresponding files—MDBR42DT.ZIP, MDBR51DT.ZIP, and MDBR81DT.ZIP—can subsequently be downloaded from the respective survey pages for 2003-04, 2008-09, and 2021.

**Funding:** The authors received no specific funding for this work.

**Competing interests:** The authors have declared that no competing interests exist.

## 1. Introduction

Neonatal mortality is defined as the occurrence of death of a newborn within 28 days of birth [1,2]. It represents a global public health burden, particularly in low- and middle-income countries. In 2020, neonatal mortality accounted for 47% of deaths in children under 5 years old, totaling 2.4 million newborns [3]. The highest rates are observed in India, sub-Saharan Africa, and central and southern Asia. At the country level, the rate ranges from 1 to 44 deaths per 1,000 live births [3]. Eighty percent of these deaths are attributable to preventable and treatable conditions, including complications related to prematurity, intrapartum factors including birth asphyxia, and neonatal infections [4]. In order to reduce neonatal mortality, cost-effective evidence-based interventions have been identified and implemented in several countries, but their impact varies from one country to another [1,2]. At the global level, neonatal mortality has decreased by 50%, or 2.4% per year from 1990 to 2021 [3,5]. However, it is slower compared to that of children aged 1–59 months, which was 3.3% per year. In sub-Saharan Africa, this reduction was approximately 39%, decreasing from 45.3 to 27.5 deaths per 1,000 live births between 1990 and 2019 [6]. Furthermore, neonatal mortality has remained stagnant for several years in certain low- and middle-income countries, starting from 1990 in the Philippines, 1994 in Ghana, 2006 in Uganda, and 2008 in Madagascar [7–13]. Potential reasons for these stagnations are multiple, including the fragility of the healthcare system, the deficiency of perinatal care coverage, and the weakness of the quality and equity of these care services, likely resulting from insufficient preparation for the implementation of cost-effective interventions. Additionally, the persistence of poverty, the behaviors and culture that are detrimental to reproductive health, as well as the persistence of early adolescent pregnancies [6–13]. Whereas, reducing neonatal mortality is a key objective of the Global Sustainable Development Goals (SDGs), aiming to achieve a mortality rate of less than 12 per 1000 live births by 2030 [6]. Many countries including Madagascar are committed to contributing to the achievement of this goal.

Although the determinants of neonatal mortality have been the subject of several studies, their temporal evolution remains poorly understood. However, the dynamics of these determinants, particularly the absence of reduction in certain factors and the emergence of new determinants, could be the cause of the observed stagnation in mortality rates. The few existing studies have mainly focused on socioeconomic and demographic factors [11,14]. They have overlooked other important aspects such as behavior and healthcare systems. Furthermore, they have presented divergent results, highlighting a large inter-country variability in the evolving trend of these factors. Given this variability and the multidisciplinary nature of the determinants of neonatal mortality, a study in a different context, which takes into account these limitations by integrating other variables for effective control of confounding factors, could enhance the understanding of the factors underlying the stagnation of neonatal mortality and contribute to its reduction.

In Madagascar, neonatal mortality remains high and stagnant, and has even shown an increase in recent years, despite the implementation of interventions

aimed at its reduction [13]. Indeed, the neonatal death rate has increased from 24 to 26 per 1000 live births between 2008 and 2021. This stagnation occurs within the context of a low-income country confronting several challenges, including underfunding of the healthcare system, a shortage of qualified personnel, disparities in healthcare access between urban and rural areas, as well as political instability and economic crises that have further strained the country's healthcare infrastructure over time. Understanding the causes behind this stagnation, through knowledge of the dynamics of the risk factors associated with neonatal mortality in the country, could contribute to the achievement of this global objective. This knowledge could guide the adaptation of existing interventions. The current study aims to investigate the dynamics of factors associated with neonatal death in Madagascar from 2003 to 2021, within the context of stagnation at a high level of neonatal mortality. Specifically, it seeks to describe the evolution of the prevalence of neonatal death by studied characteristics, identify factors associated with neonatal death, and assess the variations over time in the association of factors with neonatal mortality during the studied periods.

## 2. Methods

### 2.1. Study setting and design

We conducted a secondary analysis of cross-sectional Demographic and Health Surveys data. These surveys were carried out at a community level in Madagascar in 2003, 2008, and 2021.

Madagascar is a low-income country in sub-Saharan Africa. It is characterized by a predominantly rural population, with only 10% of this rural population belonging to the highest quintile of economic well-being. In terms of education, only 7% of women and 7% of men have completed primary school. The health system of the country is organized in a health pyramid and divided into central, regional, district, and community levels. It has experienced fluctuations in performance depending on the country's economic and political contexts. A reform initiated in the 2000s has improved access to care but was compromised in 2009 due to the economic crisis linked to political instability. Despite these efforts, the healthcare system faces persistent challenges such as underfunding, a lack of qualified personnel, inadequate infrastructure, and disparities in access to healthcare services between urban and rural areas. The healthcare system's performance before the COVID-19 pandemic was 54.5%, with difficulties in financial access [15], human resources, population utilization, and geographical accessibility of healthcare services [16]. The COVID-19 pandemic has undoubtedly exacerbated these problems and put a strain on the Malagasy healthcare system.

### 2.2. Conceptual framework of neonatal death

To analyze the complex factors that affect neonatal mortality, several conceptual frameworks have been used over time. Among the existing conceptual frameworks, the Mosley and Chen model (1984) was the first to propose a systematic approach to the determinants of infant mortality in developing countries [17]. It distinguishes between proximal and distal causes of infant mortality but does not consider social factors [18] or the various factors that lead to delays in care [19]. The main conceptual frameworks used in studies show a chronological trend, starting from a focus on the biomedical factors of neonatal mortality and gradually incorporating behavioral, social, environmental, and political dimensions.

A specific conceptual framework was developed for this study, based on that of Mosley and Chen and other similar studies [17,20–22]. Adjustments were then made based on the available data in the DHS (Fig 1).

### 2.3. Data source and sampling procedures

The current study used the Birth Recode (BR) data of the 2003, 2008, and 2021 Demographic Health Surveys (DHS) conducted in Madagascar. These surveys were carried out by the National Institute of Statistics (INSTAT) of Madagascar, and their datasets are freely available on the official website of the DHS Program [23]. These surveys initially focused on collecting information on reproductive health, mortality, nutrition, malaria, and HIV/AIDS in the country. They were conducted among women of reproductive age, including mothers of live-born children within the 5 years before

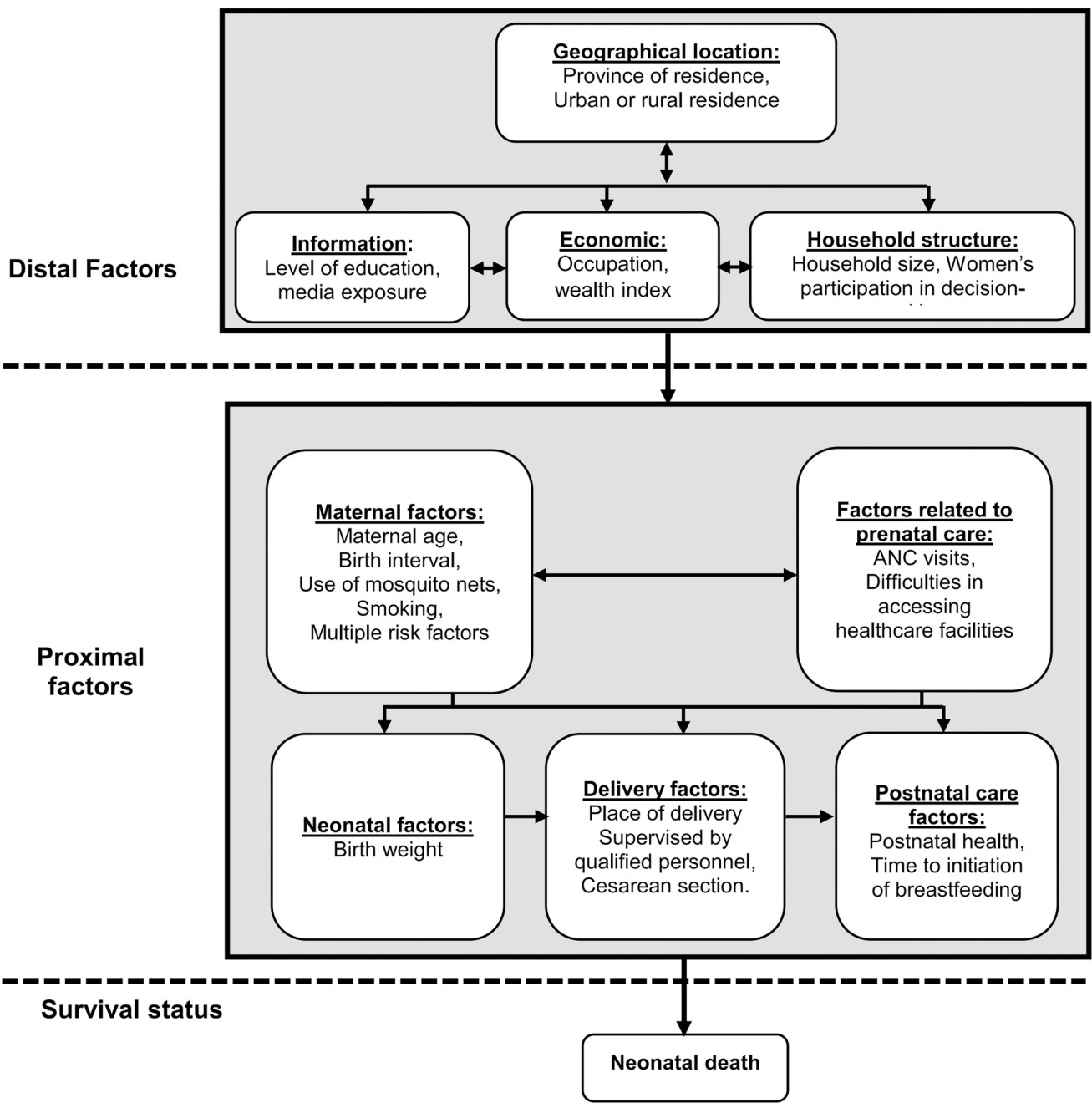

**Fig 1. Conceptual framework of neonatal mortality, adapted from models by Mosley and Chen [17].**

the survey. The sampling methods employed were based on a two-stage stratified cluster sampling design. The samples are representative at the national level, as well as at the level of the 6 provinces (in 2003), 22 regions (in 2008 and 2021), and urban and rural areas. Data were collected by teams consisting of a leader, an interviewer, and a health professional. Each member had prior experience in data collection and underwent three training sessions on the survey's specific protocols. Data collection was preceded by a pilot survey and household census. No area deemed difficult

to access was excluded, and data collection was organized by deploying teams in the country's outlying and isolated regions before concluding in the central areas and the capital, which enabled high response rates to be achieved and maintained. Data quality control measures were implemented at every stage of the data collection process to ensure data integrity. The teams from the INSTAT supervised interviews and data entry by field teams. In addition, the internal consistency of the data entered was verified. Finally, a data cleansing process was carried out, followed by a final check carried out with the help of the survey's technical team, including ICF's data processing specialist. The DHS sampling procedures for the three years are summarized in Table 1. These procedures are further detailed in the respective national reports [13,24,25].

For this study, newborns born in Madagascar during the 5 years preceding each of the 3 surveys were included. They correspond to live births during the following periods: 1998–2003, 2004–2008, and 2016–2021.

Initially, all children whose mothers were selected for the individual survey were included in the study. Then, children aged 5 years and older at the time of the survey were excluded to reduce potential recall bias, resulting in a sub-sample of children under 5 years old, which formed the final sample for this study (Fig 2). The sample sizes for children under 5 were 5,415 in 2003, 12,448 in 2008, and 12,399 in 2021.

## 2.4. Study variables

**2.4.1. Outcome variable.** The dependent variable of this study is represented by the survival status of the newborn, either alive or dead. To define cases of neonatal death, we considered deaths that occurred before the age of 30 days, a period that differs from the conventional definition of neonatal mortality (28 days). This choice was motivated by the potential lack of accuracy in determining the exact time of death for mothers who lacked documentation to support the precise date of death of their newborn. In such cases, mothers estimated the date to be one month. Furthermore, this allows for easier comparison with similar studies that also included all deaths within the first month [11,26,27].

**2.4.2. Explanatory variables.**

**Distal factors**   Distal factors encompass sociodemographic factors of the households such as the province, the area of residence (urban or rural), media exposure (the ability of the household to access sources of information such as radio, television, and the internet: yes or no), the wealth index (It is a composite measure of household living standards, constructed using principal component analysis (PCA) based on DHS methodology. PCA was applied to data encompassing household asset ownership (including television, radio, car, motorcycle, refrigerator, and telephone), dwelling characteristics (including flooring, roof, and wall materials, drinking water source, and toilet facilities), and other wealth-related indicators (including the number of rooms and agricultural land ownership). Standardized asset scores derived from the PCA were summed for each household. Individuals were then ranked based on their household's score and divided into

Table 1. Summary of the sampling procedures used in the 2003, 2008, and 2021 DHS.

| Characteristics | 2003 DHS | 2008 DHS | 2021 DHS |
|---|---|---|---|
| Level of representativeness | 6 provinces, urban and rural strata | 22 regions, urban and rural strata | 22 regions, urban and rural strata |
| Strata | 12 | 45 | 45 |
| Clusters | 300 | 596 | 650 |
| Households | 8420 | 17857 | 20510 |
| Women aged 15–49 | 7949 | 17375 | 18869 |
| Response rate | 95% | 96% | 95% |
| The data collection period | November 23, 2003 – March 28, 2004 | November 23, 2008, to August 17, 2009 | Starts on March 3, 2021, for 143 calendar days. Resumed in February 2021. |

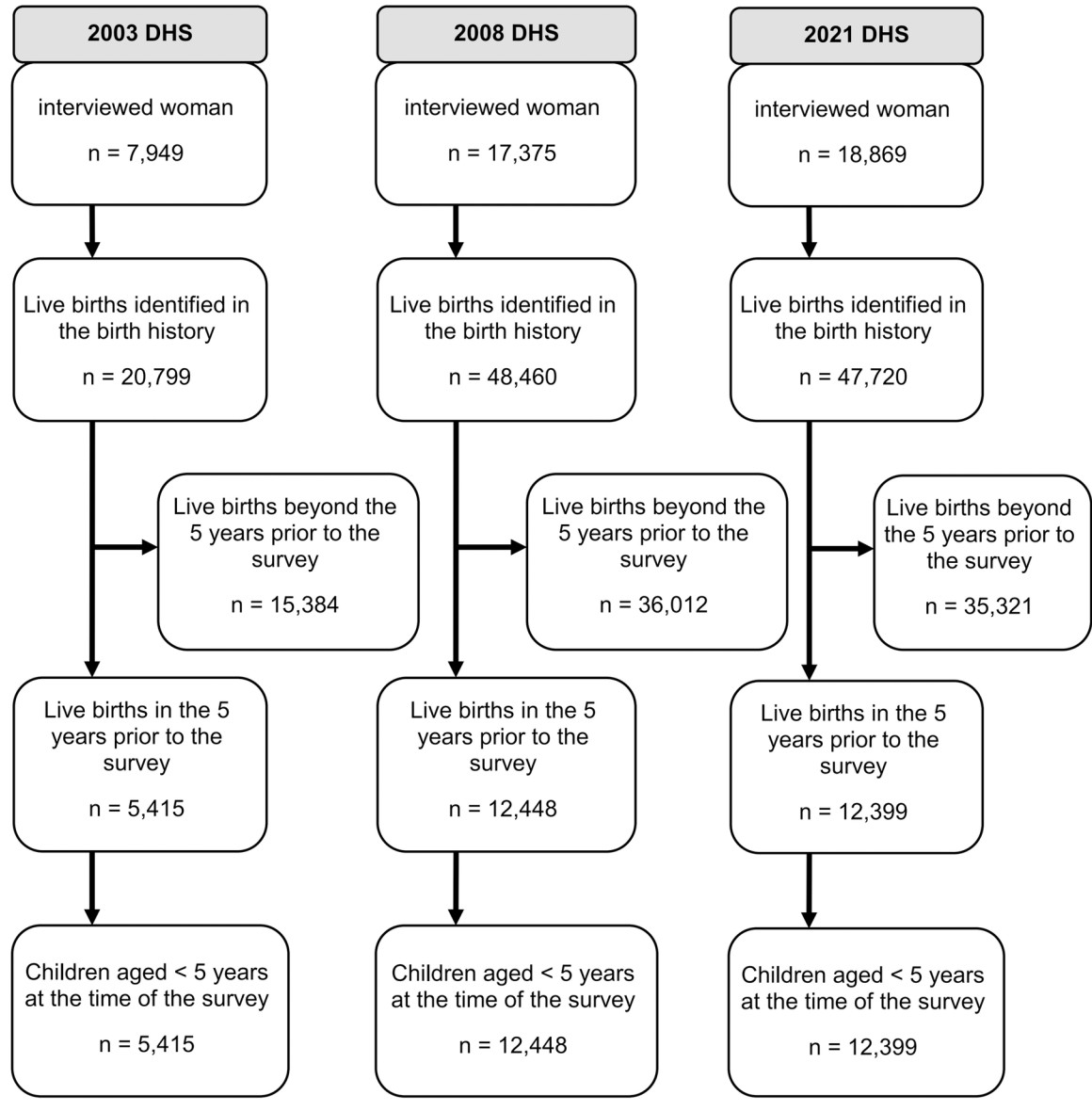

**Fig 2. Flowchart of sample selection steps in the 2003, 2008, and 2021 DHS.**

five population quintiles: poorest, poorer, middle, richer, and richest), household size (1–2, 3–5, 6 or more), mother's involvement in decision-making (It is a composite, dichotomous variable based on three DHS questions. These questions inquire about who usually makes decisions regarding their healthcare, major household purchases, and visits to family or relatives. A mother was classified as involved ("yes") if she was the sole or joint decision-maker for at least one of these questions. If none indicated her involvement, she was categorized as not involved ("no").) and the mother's occupation (yes or no).

**Proximal factors**   The proximal factors are grouped into six categories: maternal, related to prenatal care, neonatal, related to delivery, and postnatal care. Maternal factors include maternal education level (none, primary, secondary, university), maternal age at delivery (less than 20 years, 20–29 years, 30–39 years, or 40–49 years), birth intervals (less than

2 years, 2–3 years, 4 years or more), use of bed nets during pregnancy (yes or no), maternal smoking (yes or no), and the presence of multiple maternal risks (the simultaneous presence of risk factors among the following: age less than 18 years or more than 35 years, birth intervals <24 months: yes or no).

Factors related to antenatal care include the number of antenatal care (ANC) visits made by the woman during pregnancy (less than 3, 4–7, 8 or more), and geographic barriers to healthcare access (whether or not individuals experience serious difficulties in accessing healthcare for their own medical needs when sick due to the distance to healthcare facilities).

Newborn and delivery factors include birth weight (low < 2500g, normal 2500 − 4000g, high > 4000g), delivery in a health facility (yes or no), supervised delivery by skilled staff (the fact that the woman was attended by a doctor, a midwife, nurse or paramedic during delivery), and cesarean section (yes or no).

Factors related to postnatal care are represented by the newborn's pre-discharge health check (the fact that the newborn baby has been examined by a health professional before leaving the birth center), the timing of breastfeeding initiation after birth (immediate, less than 24 hours, 24 hours or more), and breastfeeding (never breastfed, breastfed).

## 2.5. Data management and analysis

Data were extracted from the Birth Recode (BR) database and analyzed using Stata 17 software (version 17, StataCorp LLC, College Station, TX 77845, USA). The dependent variable "neonatal death" was created from the quantitative variable "age of the child" expressed in years. Children aged 5 years and older were suppressed, followed by the dichotomization of the variable.

All quantitative variables were discretized to facilitate the interpretation of the results. These discretizations were performed manually using cut-offs commonly used in the literature and the DHS program statistical manual [28].

The data were declared as a survey and were weighted using the individual sample weight of women, stratification, and the primary sampling unit variable.

A descriptive analysis was carried out first. It includes a description of the sample, presenting estimates as weighted frequencies and percentages. Then, the proportion of deaths was presented for each variable and year, expressed as weighted percentages with 95% confidence intervals.

Subsequently, bivariate analyses were performed for each variable and year. Chi-square tests were applied. Crude odds ratios and their confidence intervals were calculated using simple logistic regression. Variables were selected using an ascending approach, starting with an empty model, and then gradually adding variables according to their statistical significance and their epidemiological importance in the literature. Missing values were addressed using Stata's multiple imputation methods by chained equations (ICE), with 20 imputations performed. It was assumed that the missing data occurred randomly [29]. Multiple imputed datasets were used to run multilevel logistic regressions. A post hoc sensitivity test was carried out by comparing the results of the analysis of the original data with those of the imputed data. In the end, the results of the imputed data were preferred. The selection of the multivariate model was performed with multivariate logistic regression based on several criteria, such as the Akaike information criterion (AIC) of the model, and the model's quality assessed by the Hosmer-Lemeshow test, McFadden's pseudo-R² and the area under the ROC curve (AUC). Multicollinearity was tested using the variance inflation factor (VIF) and the tolerance test. The aim was to obtain a model that included as many variables as possible while remaining statistically adequate to the data. The same explanatory variables were kept for all the years studied to track the evolution of the association measures over time while ensuring an acceptable model quality for each period [29].

Multilevel analysis was performed using a two-level multilevel logistic regression model. It is assumed that the likelihood of a child dying in the neonatal period varies depending on the community or village, as established by previous research [30]. Within the framework of the Demographic and Health Surveys (DHS), the clusters represented by enumeration areas (EAs) constitute the geographical level directly above households and are also the geographical areas closest

to the villages. In this study, the author chose to treat clusters as a community variable due to the lack of village-specific variables in the DHS. Therefore, the random effect was measured at the cluster level.

In this study, the two-level multilevel logistic regression model is written as [31]:

$$Y_{ij} = log\left(\frac{\pi_{ij}}{1 - \pi_{ij}}\right) = \gamma_{00} + \gamma_{p0}X_{pij} + \gamma_{0q}Z_{qj} + u_{oj}$$

(1)

Where i refers to the individual and j to the corresponding village. $Y_{ij}$ denotes the survival status of the $i^{th}$ child in the $j^{th}$ cluster during his neonatal period, and $\pi_{ij}$ represents the probability of neonatal death for child i in village j. $\gamma_{00}$ is the intercept of the model, representing the probability of neonatal death in the absence of explanatory variables. $X_p$ and $Z_q$ are explanatory variables at the individual and community levels, respectively. $\gamma_{p0}$ denotes the regression coefficient of the individual-level explanatory variables $X_{pij}$. $\gamma_{0q}$ is the regression coefficient of the community-level explanatory variables $X_{pij}$. Lastly, $u_{oj}$ represents the level 2 random effect.

The intraclass correlation coefficient (ICC), which represents the proportion of contextual (community-level) variance in the total variance of neonatal death, was estimated. Its value varies between 0 and 1, and a higher value suggests greater similarity between individuals within the same community, and therefore a difference between communities in terms of neonatal death. The calculation is outlined as follows:

$$ICC = \frac{\delta^2_{community}}{\left(\delta^2_{community} + \frac{\pi^2}{3}\right)}$$

(2)

Where $\delta^2_{community}$ represents the community level variance and $\frac{\pi^2}{3}$ denotes the individual level variance.

Finally, trend analyses over time were carried out by comparing the weighted proportions and adjusted odds ratios for each studied period. Improvement in the epidemiological indicators of neonatal mortality was concluded when a significant reduction in the proportion of deaths (no overlap of confidence intervals), loss of statistical significance of their association with neonatal death over time, or a progressive weakening of their positive association with neonatal death over time was observed. Conversely, deterioration of these indicators was deduced if there was a significant increase in the proportion of deaths (no overlap of confidence intervals), significant associations consistently present over time, an increase in the strength of the association, or a recent emergence of a significant positive association between the characteristic and neonatal death.

Results were reported according to the STROBE guidelines for cross-sectional studies.

## 2.6. Ethics approval and consent to participate

Although this secondary analysis meets the criteria for exempt research as defined in 45 CFR 46.102, considering that the data is publicly accessible and de-identified, approval was still obtained from the Committee for Biomedical Research Ethics (CERBM) at the Ministry of Public Health in Madagascar to conduct the study. It is important to note that the DHS databases are generated from standard DHS questionnaires, which have undergone review and approval by the institutional review board (IRB) of ICF to ensure compliance with the regulations set forth by the U.S. Department of Health and Human Services for the protection of human subjects (45 CFR 46). At the national level, the text of voluntary consent was prepared by ICF Macro based on the standard text developed by ICF Macro, and wasreviewed and approved by the CERBM at the Ministry of Public Health in Madagascar. During data collection, prior to each interview or test, written informed consent is obtained from the participant. For individuals under the age of 18, written informed consent from a parent or guardian was required and obtained. Further documentation on ethical issues related to the surveys is available at https://dhsprogram.com/methodology/Protecting-the-Privacy-of-DHS-Survey-Respondents.cfm. The data were accessed on August 31, 2022. A written authorization was obtained from The DHS Program prior to utilizing the data,

following the submission of the study protocol and agreement to adhere to the conditions of use for the DHS Program datasets. The author made no effort to search for information that could identify individual participants throughout the study process and did not have access to it.

## 3. Results

### 3.1. Description of the study population

After the selection of the population, the sample consisted of 5,415 children under 5 years old in 2003, 12,448 in 2008, and 12,399 in 2021. Fig 2 below illustrates the sample size at each stage of participant selection.

The distribution of the population and missing data according to the studied characteristics and survey years is presented in Table 2 below. Approximately a quarter of the population resides in the province of Antananarivo (27.0% in 2003, 29.0% in 2008, and 25.5% in 2021). The majority of children reside in rural areas (81.7% in 2003, 89.1% in 2008, and 84.5% in 2021). Over half of the children had a mother with primary education in 2003 and 2008 (52.0% in 2003, 55.3% in 2008). Children from the poorest households represented 28.3% of the population in 2003 and 25.8% in both 2008 and 2021. Almost half of the children lived in households with six or more people in 2003 and 2008 (49.5% in 2003, 52.3% in 2008). From 2003 to 2021, a significant proportion of children had a working mother (86.9% in 2003, 92.5% in 2008, and 86.5% in 2021).

### 3.2. Distribution and evolution of the weighted prevalence of neonatal deaths

Table 3 presents the temporal trend of the weighted prevalence of neonatal deaths from 2003 to 2021, according to the different characteristics studied. Overall, the proportion of neonatal deaths remained relatively stable over this period (3.1% [2.27-4.09] in 2003, 2.4% [2.07-2.82] in 2008, and 2.6% [2.25-2.91] in 2021). However, certain subgroups displayed notable changes in neonatal mortality rates. Specifically, neonates residing in the province of Toliara saw a significant decrease in neonatal deaths (from 4.3% [2.52-7.32] in 2003 to 1.8% [1.27-2.51] in 2021), as did those whose mothers had no education (from 4.7% [3.11-7.06] in 2003 to 2.2% [1.60-2.94] in 2021), and those who were never breastfed (60.2% [43.42-74.86] in 2003, 60.5% [50.15-69.98] in 2008, and 29.4% [23.94-35.56] in 2021). Conversely, over the same period, a significant increase in neonatal deaths was observed among newborns whose health was not checked before discharge (3.2% [2.22-4.60] in 2003, 1.8% [1.31-2.53] in 2008, and 7.6% [4.79-11.89] in 2021).

### 3.3. Factors associated with neonatal deaths

**3.3.1. Factors consistently associated with neonatal death over time.** After adjusting for potential confounding factors, a consistent association was observed between certain factors and neonatal mortality in Madagascar from 2003 to 2021 (Table 4). When other variables were held constant, it was found that low birth weight increased the odds of death during the first 30 days by 2.95 times in 2008 and by 5.52 times in 2021, compared to normal birth weight. Breastfeeding reduced the odds of death to 0.01 times that of no breastfeeding in 2003 and 2008, and to 0.05 times in 2021. Medium-sized households (composed of 3–5 people) reduced the odds of neonatal death to 0.19 times that of small households in 2008 and to 0.16 times in 2021. Similarly, large households (6 people or more) decreased the odds of death to 0.12 times that of small households in 2008 and to 0.08 times in 2021. High birth weight increased the odds of death by 2.52 times in 2008 and by 2.95 times in 2021, compared to normal birth weight.

**3.3.2. Factors associated with neonatal death that have lost their association over time.** Some factors initially found to be significantly associated with neonatal death lost this association over time (Table 4). In 2003, it was observed that children born with a birth interval of 2–3 years and 4 years or more had 0.37 and 0.35 times the odds of death, respectively, compared to those with a short birth interval, but these significant associations were no longer apparent in later years. In 2008, the odds of death for children whose mothers were aged 40–49 were 4.95 times higher compared to

**Table 2. Distribution of the population by studied characteristics.**

| Variables | 2003 | 2008 | 2021 |
|---|---|---|---|
| | N = 5,415 (%ᵃ) | N = 12,448 (%ᵃ) | N = 12,448 (%ᵃ) |
| **Province** | | | |
| Antananarivo | 1,658 (27.0) | 2,808 (29.0) | 2,740 (25.5) |
| Fianarantsoa | 984 (22.1) | 3,327 (23.3) | 2,819 (18.8) |
| Toamasina | 807 (15.4) | 1,328 (14.6) | 1,376 (13.7) |
| Mahajanga | 656 (14.7) | 1,985 (11.4) | 1,885 (12.1) |
| Toliara | 819 (13.8) | 2,245 (15.7) | 2,720 (22.3) |
| Antsiranana | 491 (6.9) | 755 (6.0) | 859 (7.5) |
| **Residence** | | | |
| Urban | 2,951 (18.4) | 2,222 (10.9) | 2,364 (15.5) |
| Rural | 2,464 (81.7) | 10,226 (89.1) | 10,035 (84.5) |
| **Wealth Index** | | | |
| Poorest | 1,092 (28.3) | 3,596 (25.8) | 3,456 (25.8) |
| Poorer | 780 (19.3) | 2,622 (22.4) | 2,758 (21.4) |
| Medium | 898 (20.9) | 2,234 (20.0) | 2,357 (20.2) |
| Richer | 1,005 (16.7) | 2,018 (17.8) | 2,013 (18.0) |
| Richest | 1,640 (14.7) | 1,978 (14.1) | 1,815 (14.7) |
| **Household size** | | | |
| 1–2 | 131 (2.3) | 243 (1.9) | 391 (3.1) |
| 3–5 | 2,696 (48.2) | 5,685 (45.8) | 6,686 (54.5) |
| >=6 | 2,588 (49.5) | 6,520 (52.3) | 5,322 (42.4) |
| **Education level** | | | |
| None | 1,127 (26.9) | 3,510 (25.4) | 3,145 (22.8) |
| Primary | 2,577 (52.0) | 6,415 (55.3) | 5,420 (44.2) |
| Secondary | 1,587 (20.2) | 2,359 (18.3) | 3,533 (30.6) |
| Academic | 124 (1.0) | 164 (1.0) | 301 (2.4) |
| **Occupation** | | | |
| Yes | 4,363 (86.9) | 11,343 (92.5) | 10,747 (86.5) |
| No | 1,052 (13.1) | 1,105 (7.5) | 1,652 (13.5) |
| **Mother's age** | | | |
| <20 years | 991 (19.1) | 2,735 (21.1) | 3,087 (25.1) |
| 20–29 years | 2,820 (51.0) | 5,967 (47.8) | 5,907 (47.5) |
| 30–39 years | 1,367 (25.2) | 3,218 (26.6) | 2,876 (23.3) |
| 40–49 years | 237 (4.7) | 528 (4.5) | 529 (4.1) |
| **Birth interval** | | | |
| <2 years | 930 (18.3) | 2,159 (17.7) | 1,952 (15.7) |
| 2 to 3 years | 2,162 (41.8) | 5,092 (41.0) | 3,811 (29.8) |
| >= 4 years | 1,002 (16.7) | 2,293 (18.7) | 3,193 (25.9) |
| Not applicable (first child) | 1,321 (23.1) | 2,904 (22.6) | 3,443 (28.6) |
| **Sleeping under a mosquito net** | | | |
| No | 3,328 (64.2) | 5,254 (45.1) | 3,931 (33.1) |
| Yes | 2,087 (35.8) | 7,194 (54.9) | 8,468 (67.0) |
| **Multiple maternal risks** | | | |
| No | 5,377 (99.1) | 12,347 (99.3) | 12,303 (99.3) |
| Yes | 38 (0.9) | 101 (0.8) | 96 (0.7) |

*(Continued)*

**Table 2.** (Continued)

| Variables | 2003 | 2008 | 2021 |
|---|---|---|---|
| | N = 5,415 (%[a]) | N = 12,448 (%[a]) | N = 12,448 (%[a]) |
| **Number of ANC[b] visits** | | | |
| <=3 | 1,849 (37.8) | 4,241 (33.9) | 3,807 (29.7) |
| 4–7 | 1,598 (25.2) | 4,040 (32.2) | 5,224 (43.2) |
| >=8 | 242 (3.2) | 288 (2.2) | 234 (2.0) |
| Unknown | 1,726 (33.9) | 3,879 (31.7) | 3,134 (25.0) |
| **Geographic barriers to healthcare access** | | | |
| Yes | 2,234 (47.4) | 2,936 (23.4) | 4,755 (36.9) |
| No | 3,181 (52.6) | 9,512 (76.6) | 7,644 (63.1) |
| **Birth weight** | | | |
| Low | 288 (13.0) | 611 (5.1) | 573 (4.9) |
| Normal | 2,030 (83.4) | 4,034 (32.8) | 3,582 (30.3) |
| High | 101 (3.5) | 7,803 (62.1) | 8,244 (64.8) |
| unweighted/unknown | 2,996 | 0 | 0 |
| **Health facility birth** | | | |
| No | 3,316 (67.6) | 7,871 (64.4) | 7,738 (61.3) |
| Yes | 2,029 (32.4) | 4,459 (35.7) | 4,602 (38.7) |
| Missing | 70 | 118 | 59 |
| **Supervised delivery by qualified personnel** | | | |
| No | 2,087 (51.8) | 5,997 (52.3) | 5,055 (38.9) |
| Yes | 3,078 (48.2) | 5,628 (47.7) | 7,344 (61.1) |
| Missing | 250 | 823 | 0 |
| **Cesarean delivery** | | | |
| No | 5,265 (99.0) | 12,263 (98.5) | 12,048 (97.5) |
| Yes | 83 (1.0) | 175 (1.5) | 291 (2.5) |
| Missing | 67 | 10 | 60 |
| **Child pre-discharge health check** | | | |
| No | 1,820 (38.3) | 2,909 (22.8) | 310 (2.8) |
| Yes | 1,488 (28.5) | 2,342 (19.3) | 3,298 (27.6) |
| Unknown/Not applicable | 2,107 (33.2) | 7,197 (57.9) | 8,791 (69.7) |
| **Breastfeeding initiation** | | | |
| Immediate | 3,503 (60.6) | 7,907 (66.5) | 7,343 (63.2) |
| <24 hrs | 1,290 (28.2) | 3,453 (26.0) | 3,567 (28.9) |
| ≥24 hrs | 549 (11.3) | 916 (7.5) | 1,111 (7.9) |
| Missing | 73 | 172 | 378 |
| **Breastfeeding** | | | |
| Never breastfed | 78 (1.7) | 216 (1.8) | 540 (4.2) |
| Breastfed | 5,280 (98.3) | 12,094 (98.2) | 11,859 (95.8) |
| Missing | 57 | 138 | 0 |

[a] Weighted Percentage;

[b] Antenatal Care

**Table 3. Weighted prevalence of neonatal death by population characteristics, Madagascar DHS.**

| Variables | 2003 | | | 2008 | | | 2021 | | |
|---|---|---|---|---|---|---|---|---|---|
| | *n* | % | [95% CI] | *n* | % | [95% CI] | *n* | % | [95% CI] |
| **Province** | | | | | | | | | |
| Antananarivo | 42 | 3.2 | [1.54-6.58] | 76 | 2.3 | [1.68-3.25] | 97 | 3.5 | [2.81-4.29] |
| Fianarantsoa | 12 | 1.4 | [0.58-3.10] | 89 | 3.1 | [2.36-3.99] | 71 | 2.7 | [2.02-3.64] |
| Toamasina | 21 | 3.6 | [2.06-6.27] | 21 | 1.6 | [1.03-2.44] | 36 | 2.7 | [1.90-3.80] |
| Mahajanga | 19 | 4.0 | [2.46-6.59] | 44 | 2.6 | [1.65-3.93] | 42 | 2.3 | [1.60-3.31] |
| Toliara | 26 | 4.3 | [2.52-7.32] | 45 | 1.9 | [1.37-2.62] | 53 | 1.8 | [1.27-2.51] |
| Antsiranana | 10 | 2.0 | [0.83-4.56] | 23 | 3.4 | [1.86-6.11] | 13 | 1.5 | [0.87-2.65] |
| **Residence** | | | | | | | | | |
| Urban | 54 | 1.8 | [1.40-2.39] | 52 | 2.6 | [1.91-3.42] | 65 | 2.8 | [2.14-3.60] |
| Rural | 76 | 3.3 | [2.40-4.61] | 246 | 2.4 | [2.03-2.84] | 247 | 2.5 | [2.17-2.91] |
| **Media exposure** | | | | | | | | | |
| Yes | 87 | 3.0 | [2.04-4.28] | 190 | 2.7 | [2.21-3.21] | 200 | 2.7 | [2.27-3.16] |
| No | 43 | 3.2 | [2.03-5.07] | 108 | 2.1 | [1.62-2.61] | 112 | 2.4 | [1.91-2.96] |
| **Mother's education level** | | | | | | | | | |
| None | 39 | 4.7 | [3.11-7.06] | 80 | 2.3 | [1.81-3.00] | 69 | 2.2 | [1.60-2.94] |
| Primary | 51 | 2.3 | [1.59-3.27] | 165 | 2.5 | [2.05-3.10] | 143 | 2.7 | [2.24-3.27] |
| Secondary | 38 | 2.9 | [1.75-4.74] | 48 | 2.2 | [1.48-3.34] | 96 | 2.7 | [2.12-3.47] |
| University | 2 | 2.1 | [0.50-8.28] | 5 | 2.5 | [1.02-6.07] | 4 | 1.5 | [0.53-3.97] |
| **Wealth index** | | | | | | | | | |
| Poorest | 34 | 3.5 | [2.28-5.20] | 81 | 2.5 | [1.89-3.37] | 72 | 2.2 | [1.68-2.98] |
| Poorer | 20 | 3.0 | [1.80-4.99] | 66 | 2.4 | [1.74-3.35] | 65 | 2.5 | [1.86-3.42] |
| Medium | 23 | 3.0 | [1.86-4.79] | 45 | 2.0 | [1.44-2.71] | 68 | 2.7 | [2.00-3.48] |
| Richer | 23 | 3.0 | [1.70-5.24] | 60 | 2.8 | [1.90-4.05] | 58 | 2.8 | [2.11-3.77] |
| Richest | 30 | 2.5 | [1.04-5.83] | 46 | 2.4 | [1.70-3.36] | 49 | 2.7 | [1.97-3.73] |
| **Household size** | | | | | | | | | |
| 1–2 | 7 | 9.3 | [4.07-19.68] | 32 | 15.4 | [9.34-24.40] | 29 | 7.9 | [5.13-11.89] |
| 3–5 | 68 | 3.3 | [2.20-5.01] | 133 | 2.3 | [1.83-2.91] | 191 | 2.9 | [2.46-3.46] |
| >= 6 | 55 | 2.5 | [1.78-3.50] | 133 | 2.0 | [1.62-2.57] | 92 | 1.7 | [1.35-2.16] |
| **Mother's participation in decision-making** | | | | | | | | | |
| No | 42 | 3.8 | [2.80-5.18] | 83 | 2.6 | [2.00-3.31] | 93 | 2.6 | [2.07-3.25] |
| Yes | 88 | 2.8 | [1.97-4.00] | 215 | 2.4 | [1.98-2.83] | 219 | 2.5 | [2.17-2.98] |
| **Mother's occupation** | | | | | | | | | |
| Yes | 107 | 3.0 | [2.24-3.90] | 263 | 2.4 | [2.01-2.76] | 267 | 2.5 | [2.17-2.88] |
| No | 23 | 3.7 | [1.76-7.57] | 35 | 3.2 | [2.09-4.84] | 45 | 2.9 | [2.06-4.16] |
| **Age of the mother** | | | | | | | | | |
| <20 years | 31 | 4.0 | [2.53-6.22] | 82 | 2.8 | [2.13-3.74] | 99 | 3.3 | [2.62-4.15] |
| 20–29 years | 58 | 2.6 | [1.74-3.73] | 122 | 2.1 | [1.63-2.57] | 120 | 2.0 | [1.59-2.51] |
| 30–39 years | 31 | 2.7 | [1.60-4.53] | 75 | 2.5 | [1.83-3.27] | 79 | 2.9 | [2.21-3.76] |
| 40–49 years | 10 | 6.6 | [2.62-15.63] | 19 | 4.3 | [2.28-7.79] | 14 | 2.6 | [1.43-4.78] |
| **Birth interval** | | | | | | | | | |
| <2 years | 30 | 4.4 | [2.83-6.63] | 54 | 2.7 | [1.96-3.68] | 52 | 2.7 | [1.92-3.75] |
| 2 to 3 years | 39 | 2.1 | [1.22-3.45] | 93 | 1.8 | [1.41-2.30] | 62 | 1.6 | [1.24-2.17] |
| >= 4 years | 17 | 2.1 | [0.78-5.31] | 45 | 2.3 | [1.59-3.37] | 79 | 2.5 | [1.93-3.31] |
| Not applicable (first child) | 44 | 4.6 | [3.28-6.27] | 106 | 3.4 | [2.70-4.31] | 119 | 3.5 | [2.82-4.26] |
| **Sleeps under a mosquito net** | | | | | | | | | |

*(Continued)*

**Table 3.** (Continued)

| Variables | 2003 | | | 2008 | | | 2021 | | |
|---|---|---|---|---|---|---|---|---|---|
| | *n* | % | [95% CI] | *n* | % | [95% CI] | *n* | % | [95% CI] |
| No | 85 | 3.0 | [2.09-4.32] | 149 | 2.8 | [2.22-3.52] | 109 | 3.0 | [2.38-3.77] |
| Yes | 45 | 3.1 | [2.08-4.67] | 149 | 2.1 | [1.71-2.59] | 203 | 2.3 | [2.01-2.72] |
| **Multiple maternal risks** | | | | | | | | | |
| No | 127 | 3.0 | [2.23-4.02] | 294 | 2.4 | [2.06-2.80] | 309 | 2.6 | [2.24-2.91] |
| Yes | 3 | 9.7 | [3.19-26.05] | 4 | 4.7 | [1.18-16.68] | 3 | 3.0 | [1.00-8.83] |
| **Number of ANC[a] visits** | | | | | | | | | |
| <3 | 39 | 2.6 | [1.73-3.86] | 71 | 1.7 | [1.22-2.23] | 83 | 2.4 | [1.90-3.05] |
| 4–7 | 17 | 2.0 | [0.93-4.12] | 57 | 1.6 | [1.15-2.20] | 78 | 1.5 | [1.19-2.00] |
| >8 | 4 | 1.2 | [0.39-3.52] | 8 | 4.8 | [2.04-10.74] | 5 | 1.9 | [0.75-4.87] |
| Unknown (not the youngest child) | 70 | 4.6 | [3.24-6.39] | 162 | 3.9 | [3.21-4.76] | 146 | 4.5 | [3.73-5.51] |
| **Geographic barriers to healthcare access** | | | | | | | | | |
| Yes | 71 | 2.9 | [1.92-4.29] | 229 | 2.5 | [2.10-2.99] | 176 | 2.3 | [1.93-2.72] |
| No | 59 | 3.3 | [2.26-4.66] | 69 | 2.1 | [1.55-2.93] | 136 | 3.0 | [2.44-3.70] |
| **Birth weight** | | | | | | | | | |
| Low | 9 | 3.4 | [1.15-9.46] | 30 | 4.0 | [2.63-6.11] | 35 | 6.2 | [4.32-8.80] |
| Normal | 14 | 0.9 | [0.37-2.02] | 58 | 1.5 | [1.06-2.05] | 50 | 1.3 | [0.93-1.81] |
| High | 1 | 0.8 | [0.11-5.19] | 210 | 2.8 | [2.32-3.35] | 227 | 2.9 | [2.47-3.33] |
| Not Weighed/Unknown | 106 | 4.1 | [3.06-5.43] | – | – | – | – | – | – |
| **Delivery in a health facility** | | | | | | | | | |
| No | 76 | 2.9 | [2.14-3.97] | 165 | 2.2 | [1.79-2.74] | 162 | 2.1 | [1.73-2.52] |
| Yes | 39 | 2.3 | [1.27-3.97] | 119 | 2.6 | [2.07-3.29] | 146 | 3.2 | [2.67-3.88] |
| **Supervised delivery by skilled personnel** | | | | | | | | | |
| No | 57 | 3.1 | [2.16-4.36] | 132 | 2.3 | [1.77-2.85] | 97 | 2.1 | [1.60-2.64] |
| Yes | 55 | 2.4 | [1.43-4.02] | 140 | 2.5 | [2.03-3.17] | 215 | 2.9 | [2.46-3.36] |
| **Cesarean delivery** | | | | | | | | | |
| No | 125 | 3.0 | [2.27-4.01] | 294 | 2.4 | [2.06-2.80] | 290 | 2.4 | [2.13-2.78] |
| Yes | 3 | 2.0 | [0.55-7.03] | 4 | 3.8 | [1.05-12.53] | 19 | 6.6 | [4.08-10.48] |
| **Health check of the baby before discharge** | | | | | | | | | |
| No | 49 | 3.2 | [2.22-4.60] | 51 | 1.8 | [1.31-2.53] | 23 | 7.6 | [4.79-11.89] |
| Yes | 28 | 2.6 | [1.64-4.19] | 22 | 1.1 | [0.65-1.69] | 53 | 1.8 | [1.35-2.49] |
| Unknown/Not Applicable | 53 | 3.3 | [1.99-5.25] | 225 | 3.1 | [2.61-3.69] | 236 | 2.6 | [2.27-3.07] |
| **Breastfeeding initiation** | | | | | | | | | |
| Immediate | 58 | 2.2 | [1.55-2.96] | 146 | 1.7 | [1.38-2.09] | 135 | 1.9 | [1.55-2.29] |
| <24h | 27 | 2.5 | [1.57-3.87] | 58 | 1.7 | [1.26-2.32] | 60 | 1.4 | [1.03-1.94] |
| >24h | 12 | 2.5 | [1.28-4.90] | 20 | 2.4 | [1.15-4.78] | 26 | 2.5 | [1.60-3.88] |
| **Breastfeeding** | | | | | | | | | |
| Never Breastfed | 42 | 60.2 | [43.42-74.86] | 128 | 60.5 | [50.15-69.98] | 142 | 29.4 | [23.94-35.56] |
| Breastfed | 73 | 1.7 | [1.21-2.45] | 154 | 1.2 | [0.99-1.52] | 170 | 1.4 | [1.16-1.66] |

[a] Antenatal Care

**Table 4.** Multilevel analysis of factors associated with neonatal deaths, 2003, 2008, and 2021 Madagascar DHS.

| Variables | 2003 | | 2008 | | 2021 | |
|---|---|---|---|---|---|---|
| | OR[a] [95% CI] | AOR[b] [95% CI] | OR[a] [95% CI] | AOR[b] [95% CI] | OR[a] [95% CI] | AOR[b] [95% CI] |
| **FIXED EFFECTS** | | | | | | |
| **HOUSEHOLD CHARACTERISTICS** | | | | | | |
| Province | | | | | | |
| Antananarivo | 1 | 1 | 1 | 1 | 1 | 1 |
| Fianarantsoa | 0.41 [0.13,1.27] | 0.43 [0.11,1.73] | 1.32 [0.86,2.04] | 1.60 [0.87,2.96] | 0.78 [0.53,1.13] | 1.20 [0.68,2.10] |
| Toamasina | 1.13 [0.44,2.89] | 1.40 [0.53,3.67] | 0.67 [0.39,1.17] | 1.01 [0.44,2.31] | 0.77 [0.51,1.17] | 0.52 [0.26,1.01] |
| Mahajanga | 1.27 [0.51,3.13] | 0.86 [0.22,3.44] | 1.09 [0.62,1.91] | 1.69 [0.84,3.38] | 0.66 [0.43,1.01] | 0.70 [0.37,1.35] |
| Toliara | 1.36 [0.54,3.44] | 1.29 [0.47,3.51] | 0.81 [0.50,1.29] | 0.93 [0.42,2.07] | 0.51 [0.34,0.76] ** | 0.40 [0.23,0.70] ** |
| Antsiranana | 0.60 [0.19,1.88] | 0.87 [0.21,3.51] | 1.47 [0.73,2.96] | 0.64 [0.19,2.20] | 0.43 [0.23,0.79] ** | 0.42 [0.17,1.01] |
| **Residence** | | | | | | |
| Urban | 1 | 1 | 1 | 1 | 1 | 1 |
| Rural | 1.85 [1.20,2.85] ** | 1.40 [0.63,3.08] | 0.94 [0.66,1.32] | 0.99 [0.51,1.93] | 0.90 [0.66,1.23] | 0.71 [0.45,1.10] |
| **Media exposure** | | | | | | |
| Yes | 1 | 1 | 1 | 1 | 1 | 1 |
| No | 1.09 [0.60,1.96] | 0.61 [0.27,1.37] | 0.77 [0.57,1.03] | 0.70 [0.47,1.04] | 0.88 [0.66,1.18] | 0.94 [0.60,1.48] |
| **Mother's education level** | | | | | | |
| None | 2.31 [0.52,10.21] | 2.65 [0.22,32.70] | 0.92 [0.36,2.39] | 3.88 [1.04,14.41] * | 1.49 [0.51,4.35] | 2.77 [0.70,11.00] |
| Primary | 1.09 [0.26,4.68] | 0.96 [0.09,10.63] | 1.00 [0.39,2.55] | 3.80 [1.10,13.14] * | 1.87 [0.66,5.32] | 2.57 [0.70,9.49] |
| Secondary | 1.39 [0.32,6.01] | 0.90 [0.08,10.19] | 0.88 [0.33,2.39] | 4.09 [1.18,14.20] * | 1.87 [0.65,5.37] | 2.87 [0.81,10.22] |
| University | 1 | 1 | 1 | 1 | 1 | 1 |
| **Household size** | | | | | | |
| 1–2 | 1 | 1 | 1 | 1 | 1 | 1 |
| 3–5 | 0.34 [0.13,0.90] * | 0.81 [0.14,4.53] | 0.13 [0.07,0.24] *** | 0.19 [0.05,0.67] * | 0.35 [0.22,0.57] *** | 0.16 [0.08,0.33] *** |
| >=6 | 0.25 [0.09,0.67] ** | 0.61 [0.10,3.70] | 0.11 [0.06,0.21] *** | 0.12 [0.03,0.41] *** | 0.20 [0.12,0.35] *** | 0.08 [0.04,0.17] *** |
| **Mother's Participation in Decision-Making** | | | | | | |
| No | 1 | 1 | 1 | 1 | 1 | 1 |
| Yes | 0.73 [0.49,1.08] | 0.63 [0.36,1.12] | 0.92 [0.68,1.24] | 1.05 [0.69,1.59] | 0.98 [0.74,1.30] | 1.25 [0.84,1.87] |
| **Mother's occupation** | | | | | | |
| Yes | 1 | 1 | 1 | 1 | 1 | 1 |
| No | 1.25 [0.62,2.55] | 1.72 [0.74,3.97] | 1.37 [0.87,2.13] | 1.57 [0.75,3.30] | 1.18 [0.79,1.75] | 1.19 [0.70,2.02] |
| **MOTHER'S CHARACTERISTICS** | | | | | | |
| **Mother's age** | | | | | | |
| <20 years | 1 | 1 | 1 | 1 | 1 | 1 |
| 20–29 years | 0.63 [0.34,1.16] | 1.65 [0.50,5.44] | 0.72 [0.51,1.02] | 1.40 [0.76,2.60] | 0.60 [0.42,0.84] ** | 0.95 [0.60,1.52] |
| 30–39 years | 0.67 [0.33,1.37] | 2.34 [0.62,8.83] | 0.86 [0.56,1.33] | 2.25 [0.90,5.62] | 0.87 [0.60,1.26] | 1.68 [0.87,3.26] |
| 40–49 years | 1.70 [0.68,4.28] | 2.82 [0.31,25.43] | 1.53 [0.76,3.08] | 4.95 [1.54,15.93] ** | 0.79 [0.40,1.56] | 1.82 [0.56,5.86] |
| **Birth interval** | | | | | | |
| <2 years | 1 | 1 | 1 | 1 | 1 | 1 |
| 2 to 3 years | 0.46 [0.24,0.91] * | 0.37 [0.17,0.81] * | 0.66 [0.45,0.98] * | 0.74 [0.46,1.21] | 0.60 [0.40,0.92] * | 0.83 [0.49,1.41] |
| >= 4 years | 0.46 [0.17,1.27] | 0.35 [0.14,0.88] * | 0.86 [0.52,1.42] | 0.53 [0.27,1.03] | 0.94 [0.60,1.47] | 0.92 [0.51,1.65] |
| Not applicable (first child) | 1.05 [0.63,1.75] | 1.29 [0.44,3.81] | 1.28 [0.88,1.87] | 1.09 [0.57,2.08] | 1.30 [0.86,1.96] | 1.31 [0.77,2.21] |

*(Continued)*

| Variables | 2003 | | 2008 | | 2021 | |
|---|---|---|---|---|---|---|
| | ORᵃ [95% CI] | AORᵇ [95% CI] | ORᵃ [95% CI] | AORᵇ [95% CI] | ORᵃ [95% CI] | AORᵇ [95% CI] |
| **Sleeps Under a Mosquito Net** | | | | | | |
| No | 1 | 1 | 1 | 1 | 1 | 1 |
| Yes | 1.04 [0.63,1.72] | 0.97 [0.49,1.91] | 0.75 [0.54,1.03] | 0.47 [0.29,0.74] ** | 0.78 [0.58,1.03] | 0.86 [0.58,1.27] |
| **Multiple maternal risks** | | | | | | |
| No | 1 | 1 | 1 | 1 | 1 | 1 |
| Yes | 3.49 [1.12,10.93] * | 1.64 [0.17,15.97] | 1.99 [0.49,8.15] | 1.37 [0.30,6.21] | 1.19 [0.38,3.72] | 2.38 [0.43,13.10] |
| **PRENATAL CARE** | | | | | | |
| **Number of ANCᶜ Visits** | | | | | | |
| <3 | 1 | 1 | 1 | 1 | 1 | 1 |
| 4–7 | 0.75 [0.35,1.63] | 0.70 [0.25,1.97] | 0.96 [0.62,1.49] | 1.07 [0.59,1.95] | 0.63 [0.44,0.92] * | 0.38 [0.20,0.70] ** |
| >8 | 0.45 [0.14,1.47] | 1.15 [0.24,5.56] | 2.97 [1.17,7.53] * | 0.38 [0.06,2.65] | 0.79 [0.29,2.14] | 0.39 [0.05,2.96] |
| Unknown (not the youngest child) | 1.80 [1.13,2.88] * | 3.08 [1.57,6.04] ** | 2.42 [1.69,3.46] *** | 2.82 [1.49,5.33] ** | 1.92 [1.42,2.61] *** | 3.28 [1.95,5.51] *** |
| **Geographic barriers to healthcare access** | | | | | | |
| Yes | 1 | 1 | 1 | 1 | 1 | 1 |
| No | 1.13 [0.68,1.88] | 0.86 [0.43,1.73] | 0.85 [0.58,1.23] | 0.83 [0.52,1.32] | 1.32 [0.99,1.75] | 1.46 [1.01,2.10] * |
| **NEWBORN AND DELIVERY CHARACTERISTICS** | | | | | | |
| Birth weight | | | | | | |
| Low | 3.96 [1.65,9.54] ** | 2.90 [0.61,13.72] | 2.81 [1.63,4.82] *** | 2.95 [1.42,6.12] ** | 5.02 [3.02,8.33] *** | 5.52 [2.98,10.21] *** |
| Normal | 1 | 1 | 1 | 1 | 1 | 1 |
| High | 0.87 [0.10,7.32] | 1.26 [0.07,21.34] | 1.92 [1.32,2.79] *** | 2.52 [1.40,4.56] ** | 2.24 [1.57,3.21] *** | 2.95 [1.75,4.98] *** |
| Not weighed/unknown | 4.83 [1.99,11.74] *** | 4.91 [1.34,18.06] * | | | | |
| **Supervised delivery by skilled Personnel** | | | | | | |
| No | 1 | 1 | 1 | 1 | 1 | 1 |
| Yes | 0.78 [0.40,1.50] | 1.38 [0.54,3.54] | 1.13 [0.81,1.59] | 1.34 [0.75,2.38] | 1.41 [1.03,1.93] * | 1.59 [1.04,2.44] * |
| **Cesarean delivery** | | | | | | |
| No | 1 | 1 | 1 | 1 | 1 | 1 |
| Yes | 0.65 [0.17,2.46] | 1.23 [0.20,7.62] | 1.59 [0.43,5.89] | 1 | 2.83 [1.69,4.75] *** | 1.30 [0.51,3.33] |
| **POSTNATAL CARE** | | | | | | |
| **Health check of the baby before discharge** | | | | | | |
| No | 1 | 1 | 1 | 1 | 1 | 1 |
| Yes | 0.82 [0.47,1.41] | 1.36 [0.68,2.69] | 0.57 [0.34,0.96] * | 0.65 [0.34,1.27] | 0.23 [0.12,0.41] *** | 0.71 [0.22,2.32] |
| Unknown/Not applicable | 1.01 [0.56,1.84] | 1.32 [0.45,3.86] | 1.73 [1.19,2.51] ** | 0.75 [0.37,1.53] | 0.33 [0.20,0.56] *** | 0.31 [0.08,1.15] |
| **Breastfeeding initiation** | | | | | | |
| Immediate | 1 | 1 | 1 | 1 | 1 | 1 |
| <24h | 1.15 [0.77,1.73] | 1.02 [0.57,1.81] | 1.01 [0.71,1.44] | 0.72 [0.47,1.11] | 0.75 [0.52,1.08] | 0.71 [0.46,1.12] |
| >24h | 1.18 [0.58,2.42] | 0.79 [0.21,2.95] | 1.40 [0.65,3.00] | 0.99 [0.45,2.17] | 1.33 [0.82,2.16] | 1.10 [0.66,1.84] |
| **Breastfeeding** | | | | | | |
| Never breastfed | 1 | | 1 | 1 | 1 | 1 |
| Breastfed | 0.01 [0.01,0.02] *** | | 0.01 [0.01,0.01] *** | 0.01 [0.00,0.01] *** | 0.03 [0.02,0.05] *** | 0.05 [0.03,0.09] *** |

*(Continued)*

**Table 4.** (Continued)

| Variables | 2003 | | 2008 | | 2021 | |
|---|---|---|---|---|---|---|
| | OR[a] [95% CI] | AOR[b] [95% CI] | OR[a] [95% CI] | AOR[b] [95% CI] | OR[a] [95% CI] | AOR[b] [95% CI] |
| **Random effects** | | | | | | |
| Cluster variance | | 1.97 [0.87,3.08] *** | | 1.13 [0.58,1.67] *** | | 0.59 [0.22,0.96] ** |
| PCV[d] | | -64.60 | | -68.71 | | -25.26 |
| ICC[e] | | 37.47 | | 25.53 | | 15.31 |
| MOR[f] | | 3.75 [2.32-5.18] | | 2.75 [2.08-3.43] | | 2.09 [1.61-2.56] |
| **Model statistics** | | | | | | |
| AIC[g] | | 1015.33 | | 1582.20 | | 1719.97 |
| Hosmer-Lemeshow (chi2; p-value) | | 6.77, p=0.562 | | 4.33, p=0.826 | | 7.58, p=0.476 |
| HATSQ (linktest) | | -0.05, p=0.963 | | 0.53, p=0.593 | | 1.27, p=0.204 |
| AUC[h] (%) | | 82.07 | | 80.78 | | 82.48 |

*p-value<0.05;

**p-value<0.01;

***p-value<0.001;

[a] odds-ratio;

[b] adjusted odds-ratio;

[c] Antenatal Care;

[d] Proportional Change in Variance;

[e] Intraclass Correlation Coefficient;

[f] MOR: Median Odds Ratio;

[h] AUC: Area Under the ROC Curve

those whose mothers were under 20 years old, but this significant association was not observed in 2021. Maternal use of a mosquito net in 2008 was associated with 0.47 times the odds of neonatal death compared to no mosquito net use that year. However, this effect was not observed in 2021.

**3.3.3. Factors recently associated with neonatal death.** Over time, new factors associated with neonatal death have emerged, as outlined in Table 4. In 2021, it was observed that children residing in the province of Toliara had 0.40 times the odds of neonatal death compared to those living in the province of Antananarivo. Children whose mothers attended 4–7 ANC visits had 0.38 times the odds of death compared to those whose mothers attended less than 3 ANC visits. In 2021, children in households facing no geographic barriers to healthcare access had 1.46 times higher odds of death during the neonatal period compared to those encountering such barriers. It was also observed that supervised delivery was associated with 1.59 times higher odds of neonatal death compared to non-supervised delivery in 2021.

**3.3.4. Results of random-effects measurements and model statistics.** A significant cluster-level variance of neonatal death was observed over different periods, showing a decreasing trend from 2003 to 2021 (1.97 [0.87, 3.08] in 2003, 1.13 [0.58, 1.67] in 2008, and 0.59 [0.22, 0.96] in 2021) as shown in Table 4. The intracluster correlation coefficient (ICC) also decreased, from 37.5% in 2003 to 25.5% in 2008, and further to 15.3% in 2021. Individual and community factors explained 96.2%, 85.8%, and 82.7% of the variance of neonatal death in 2003, 2008, and 2021, respectively. The median odds ratio (MOR) followed the same decreasing trend, from 3.75 (95% CI: 2.32-5.18) in 2003 to 2.75 (95% CI: 2.08-3.43) in 2008, and then to 2.09 (95% CI: 1.61-2.56) in 2021. These significant values indicate that the multilevel analysis is useful for each of the models. Furthermore, the decreases suggest a reduction in disparities of neonatal death across clusters over the years. The Hosmer-Lemeshow test showed a good fit of the model for all years, with non-significant p-values (p=0.562 in 2003, p=0.826 in 2008, and p=0.476 in 2021). The link function test (Hatsq) yielded

non-significant results with values of (-0.05, p = 0.963) in 2003, (0.53, p = 0.593) in 2008, and (1.27, p = 0.204) in 2021, indicating no model specification problem in the three periods studied. Finally, the area under the curve (AUC) was 82.1% in 2003, 80.8% in 2008, and 82.5% in 2021, demonstrating a high and stable discriminatory capacity of the model over the years.

## 4. Discussion

The present study aims to investigate the dynamics of factors associated with neonatal mortality in Madagascar between 2003 and 2021 in the context of stagnant neonatal mortality rates. Our findings showed a stagnation in the prevalence of neonatal deaths from 2003 to 2021. Factors associated with improvements in epidemiological indicators of neonatal mortality over time included province of residence, age and education level of the mother, birth intervals, mosquito net use, number of ANC visits, and breastfeeding. On the other hand, factors such as household size, geographic barriers to healthcare access, low birth weight, supervised delivery by skilled personnel and child health check before discharge were associated with worsening neonatal mortality indicators over time.

### 4.1. Neonatal mortality trends in Madagascar

The evolution of neonatal mortality in Madagascar remained stagnant from 2003 to 2021, with a slight non-significant decrease between 2003 and 2008. This trend reflects that observed in sub-Saharan Africa, the Middle East, North Africa, and low-income countries, where stagnation began later, around 2010, after a period of more rapid reduction [6]. In contrast, at the global level, the reduction remained significant from 1999 to 2019. The slight improvement observed in Madagascar between 2003 and 2008 coincides with the implementation and strengthening of preventive community interventions in various regions of the country during this period [16]. This phenomenon is in line with the improvements observed globally and in sub-Saharan Africa before 2010, attributed to the implementation of high-impact interventions such as peripartum and postnatal care, nutrition strengthening, vaccination, and hygiene since 1990 [6]. However, the impact of these interventions seems to be less pronounced in Madagascar than in other regions. The post-2010 stagnation identified in West and Central Africa, the Middle East, North Africa, and the average of low-income countries could be due to the more pronounced persistence of the common structural deficits mentioned by Sharrow et al. as a cause of the slow global reduction in neonatal deaths [6].

### 4.2. Characteristics associated with improvement in epidemiological indicators of neonatal mortality over time

Residing in the Toliara province, birth interval greater than 2 years, 4–7 ANC visits, use of mosquito net, and the absence of breastfeeding were associated with a lower risk of neonatal death. In addition, the mother's lack of education and her age (40–49) increased the risk of neonatal death. These factors showed improvement in neonatal mortality indicators (prevalence and adjusted odds ratio) over time. The results of the random effects of neonatal deaths between clusters have also highlighted a reduction in the heterogeneity of neonatal mortality between clusters over time, and could probably be linked to improvements within the previous associated factors.

The association between neonatal mortality and each of these factors are generally consistent with the majority of studies. Some authors have found that high rates of neonatal mortality differ by province due to regional variations in the implementation of health policies [26,32]. In the case of Toliara province, the reduction in neonatal mortality could be attributed to the convergence of several national and provincial factors operating at the community level. These include improved access due to the rehabilitation of the national road in 2004, the impact of decentralization and deconcentration processes since 2004 (particularly benefiting rural areas), the expansion of vaccination coverage, and the promotion of maternal breastfeeding in rural and remote areas. The probable impact of implementing the 2005–2015 Roadmap for the Reduction of Maternal and Neonatal Mortality in Madagascar, which aimed to strengthen community-based interventions, should also be considered. Furthermore, the observed shift in the urban-rural prevalence of neonatal mortality, with a

trend towards higher prevalence in urban areas, warrants further consideration. The increased prevalence of neonatal mortality in urban settings, particularly within the capital, Antananarivo, may reflect challenges associated with urbanization. These could include factors such as increased population density, disparities in living conditions and access to resources within the urban environment, and potentially, shifts in health behaviors or social support networks.

The mother's level of education is particularly important for improving knowledge of her child's health and use of health services, unlike the father's, which generally facilitates access to well-paid jobs [33]. Children born with short birth intervals were conceived under unfavorable maternal health conditions, and were exposed to pregnancy complications due to inadequate recovery after the previous pregnancy [34]. Pregnancies in elderly mothers present a high risk of gestational diabetes and gestational hypertension. Uteroplacental aging can also induce circulatory insufficiency and placental hypoperfusion, compromising fetal growth [35]. The absence of protection by a mosquito net exposes mothers to malaria-related complications during pregnancy [36–38]. Antenatal care (ANC) reinforces maternal education and compliance, while also offering screening for pregnancy complications, infection treatment, and early management of newborn illnesses [39]. Non-breastfed infants are deprived of the benefits of breastfeeding, exposing them to risks of dehydration, malnutrition, or infections [40].

The improvements observed for each of these characteristics could be attributed to the impact of increased coverage of preventive and promotional services particularly for elderly mothers. This expanded coverage includes increased use of modern contraceptives associated with a reduction in unmet family planning needs [41], wider distribution and use of mosquito nets, improved attendance of adequate antenatal care visits, and intensified large-scale promotion of breastfeeding at the community level [42].

## 4.3. Characteristics associated with worsening of epidemiological indicators of neonatal mortality over time

Ease of access to health centers, low and high birth weight, supervised delivery by skilled personnel, and child health check before discharge were associated with an increased risk of mortality during the neonatal period. Furthermore, medium and large household size were found to decrease the risk of neonatal death. For each of these factors, a deterioration in epidemiological (prevalence and adjusted odds ratio) indicators of neonatal mortality was observed over the years.

Among these factors, only the association of birth weight anomalies with neonatal death aligns with existing evidence. Children with extreme birth weights are more likely to develop respiratory, infectious, or metabolic complications due to their increased vulnerability [39,43]. The absence of a child health check-up at discharge was not significantly associated with neonatal mortality in this study, although it is a factor that can lead to delays in the diagnosis and treatment of neonatal complications, such as infections, asphyxia or hypothermia [44]. Existing evidence shows that small household size, easy access to health-care facilities and supervised delivery by qualified personnel contribute to reducing neonatal mortality, which is not in line with our result. The divergence of our results could be explained by several reasons. According to some authors, a low probability of neonatal death in larger households could be explained by the availability of an experienced person capable of taking care of the mother and newborn [45]. Therefore, high mortality in smaller households could indicate parental incompetence in adequately caring for sick or small newborns. Numerous studies have shown that the presence of qualified personnel during childbirth is a protective factor against neonatal mortality [46,47], although the results vary by region. In Asia, the presence of qualified attendants during childbirth reduced neonatal mortality only during the first week of life, while in Africa, it is associated with higher neonatal mortality due to the habit of the majority of the population to seek qualified assistance only in case of complications, which could bias the effect of these qualified personnel. This finding also highlights gaps in the training and equipment of this personnel [48]. Our results are consistent with the findings of Singh and could be explained by his hypotheses.

The deteriorations in neonatal mortality indicators observed over time could be explained by deficiencies in both delivery and postnatal care, worsening over time. In other words, for birth weight anomalies, it could reveal the inefficiency of

interventions aimed at preventing and managing preterm births, intrauterine growth retardation, and complications associated with extreme birth weights. The absence of child health verification upon discharge could indicate the persistence of poor delivery conditions, lack of medical follow-up, early discharge without consent, or a shortage of healthcare personnel in maternal and child health [49]. Regarding small household size, it could demonstrate the persistence of insufficient social and educational support for parents unable to adequately care for their sick or weak newborns. Regarding ease of access to healthcare centers, particularly in urban areas, the deterioration could reflect insufficient quality of care provided to the mother and child. The coronavirus pandemic would also likely have influenced the study results in urban settings. Finally, regarding supervised delivery by skilled personnel, the deterioration in neonatal mortality indicators over time would indicate and underline the growing need to improve the quality of birth attendant training and equipment, as well as access to emergency obstetric care.

## 4.4. Limitations and validity of the study

This study provides valuable insights into neonatal mortality trends across Madagascar's 22 regions, based on nationally representative data. However, several limitations should be acknowledged. The cross-sectional nature of the survey data, which relies on retrospective maternal reports, introduces potential for both recall bias and misclassification of neonatal deaths. Mothers may face challenges in accurately remembering the precise timing of death, particularly for events from the more distant past or those occurring during emotionally stressful periods. This difficulty is compounded by the absence of formal records, like health cards or death certificates, which are less common for births outside health facilities, often assisted by traditional birth attendants. Consequently, distinguishing between a stillbirth and an early neonatal death becomes more difficult, potentially leading to misclassification. Furthermore, underreporting of such births could contribute to an underestimation of neonatal mortality, especially in rural areas. Given the nature of the data, the quality is paramount and may influence the findings, potentially elucidating the observed territorial variations. The potential for underreporting and misclassification should therefore be considered when interpreting the observed disparities, including the seemingly elevated mortality rates in populations typically deemed to be at lower risk. Moreover, the quality of data collection methods may have improved over time, which could impact the observed trends.

It should also be noted that the data used do not allow for the evaluation of the effectiveness and efficiency of existing or potential interventions to reduce neonatal mortality, nor do they measure the impact of maternal and child health policies and programs. Evaluation studies based on randomized trials or cost-effectiveness analyses could be relevant to identify best practices and optimal strategies for reducing neonatal mortality.

Furthermore, the analyses could be biased by missing data. To manage the impact of these missing data, some were categorized as "unknown" or "not applicable" when part of the population was not concerned by the question asked in the variable, while the remaining missing data were treated by multiple imputation.

Another limitation is that the data used do not allow for the identification of the exact causes of these deaths, such as neonatal infections, asphyxia, prematurity, or congenital malformation. Some risk factors that are commonly associated with neonatal mortality were not examined in this study due to their unavailability in the DHS data, which may limit the study's scope. These unexamined factors include the rates of prematurity, intrauterine growth restriction, and maternal health conditions such as gestational diabetes and HIV. Future research should take into account these factors to overcome this limitation.

Additionally, political and environmental factors such as governance and the performance of the health system were not taken into account due to their unavailability in the DHS database. However, the results regarding factors related to geographic barriers to healthcare access and the quality of care in this study may reflect these unaccounted factors.

The study consolidated data on childbirth assistance into a dichotomous variable "assisted delivery by qualified healthcare personnel" (yes/no), although information regarding traditional birth attendants was available. This simplification aimed to enhance statistical power, facilitate comparison with other studies, and focus on the primary objective of

identifying neonatal mortality factors from a broad perspective. Nevertheless, this methodological approach does not allow for observation of the potential role of traditional birth attendants in neonatal mortality. We acknowledge this limitation of the study. A comprehensive analysis of the impact of these practitioners would represent a relevant avenue for future research.

## 5. Conclusion

Overall, prenatal prevention and care have generally been associated with improved neonatal mortality indicators, while care during delivery and the postnatal period has been associated with a deterioration of these indicators. This study also found that the effect of the quality of neonatal care in available health facilities on neonatal death is greater than that of geographical accessibility.

Our recommendations for Madagascar underscore several critical priorities. It is imperative to concentrate efforts on enhancing the quality of care provided during childbirth and the postnatal period in easily accessible health centers, particularly within urban areas. This specifically entails upgrading equipment and care protocols for both low- and high-birth-weight newborns, re-evaluating and updating the training of skilled birth attendants, and improving the quality of equipment utilized by these professionals. Furthermore, it is essential to strengthen protocols for assessing the health of newborns before hospital discharge, develop targeted education and support programs for parents in small households, and establish community support systems to assist these families in caring for their newborns at home.

In the future, it would be important to evaluate, using rigorous methods, the impact of community interventions on reducing neonatal mortality in Madagascar. In addition, it would be pertinent to evaluate the effectiveness and impact of targeted interventions aimed at improving the care of low-birth-weight newborns, as well as the competencies of parents from small households in providing home-based care for their newborns. An in-depth study of the quality of obstetric and neonatal care covering all regions and districts of the country would also be of great importance, particularly during the postpartum period. Finally, an analysis of parents' knowledge, attitudes and practices regarding home neonatal care is essential.

## Acknowledgments

The research team would like to thank the DHS program for approving the use of the Madagascar DHS database for this study.

## Author contributions

**Conceptualization:** Sedera Radoniaina Rakotondrasoa, Kadari Cissé.

**Data curation:** Sedera Radoniaina Rakotondrasoa.

**Formal analysis:** Sedera Radoniaina Rakotondrasoa.

**Methodology:** Sedera Radoniaina Rakotondrasoa, Kadari Cissé, Tieba Millogo, Hajalalaina Rabarisoa, Felix Alain, Seni Kouanda, Julio El-C Rakotonirina.

**Project administration:** Sedera Radoniaina Rakotondrasoa.

**Resources:** Sedera Radoniaina Rakotondrasoa.

**Supervision:** Kadari Cissé, Tieba Millogo, Seni Kouanda, Julio El-C Rakotonirina.

**Validation:** Kadari Cissé, Tieba Millogo, Hajalalaina Rabarisoa, Felix Alain, Seni Kouanda, Julio El-C Rakotonirina.

**Visualization:** Sedera Radoniaina Rakotondrasoa.

**Writing – original draft:** Sedera Radoniaina Rakotondrasoa, Hajalalaina Rabarisoa, Felix Alain.

**Writing – review & editing:** Sedera Radoniaina Rakotondrasoa, Kadari Cissé, Tieba Millogo, Hajalalaina Rabarisoa, Felix Alain, Seni Kouanda, Julio El-C Rakotonirina.

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
