## [Decision Letter · Decision Letter 0]

20 Nov 2024

PGPH-D-24-02012

“Dynamics of factors associated with neonatal death in Madagascar: a comparative analysis of the 2003, 2008, 2021 DHS”

Dear Dr. Rakotondrasoa,

Thank you for submitting your manuscript to PLOS Global Public Health. After careful consideration, we feel that it has merit but does not fully meet PLOS Global Public Health’s publication criteria as it currently stands. Therefore, we invite you to submit a revised version of the manuscript that addresses the points raised during the review process.

Please address the clarifications requested by the second reviewer has recommended in relation to the introduction and methodology. In addition, please add more explanation on:

1- how 'mother’s involvement in decision-making' is defined and categorised.

2- how wealth index constructed and what is included in the index.

We look forward to receiving your revised manuscript.

Kind regards,

Hassan Haghparast Bidgoli

Academic Editor

Journal Requirements:

1. We do not publish any copyright or trademark symbols that usually accompany proprietary names, eg (R), (C), or TM  (e.g. next to drug or reagent names). Please remove all instances of trademark/copyright symbols throughout the text, including Stata 17® on page 9 and 12.

2. Please provide an Author Summary. This should appear in your manuscript between the Abstract (if applicable) and the Introduction, and should be 150–200 words long. The aim should be to make your findings accessible to a wide audience that includes both scientists and non-scientists. Sample summaries can be found on our website under Submission Guidelines: 

https://journals.plos.org/globalpublichealth/s/submission-guidelines#loc-parts-of-a-submission

Additional Editor Comments (if provided):

Please address the clarifications requested by the second reviewer has recommended in relation to the introduction and methodology. In addition, please add more explanation on:

1- how 'mother’s involvement in decision-making' is defined and categorised.

2- how wealth index constructed and what is included in the index.

Reviewers' comments:

Reviewer's Responses to Questions

**Comments to the Author**

1. Does this manuscript meet PLOS Global Public Health’s publication criteria ? Is the manuscript technically sound, and do the data support the conclusions? The manuscript must describe methodologically and ethically rigorous research with conclusions that are appropriately drawn based on the data presented.

Reviewer #1: Yes

Reviewer #2: Yes

2. Has the statistical analysis been performed appropriately and rigorously?

Reviewer #1: Yes

Reviewer #2: Yes

3. Have the authors made all data underlying the findings in their manuscript fully available (please refer to the Data Availability Statement at the start of the manuscript PDF file)?

Reviewer #1: Yes

Reviewer #2: Yes

4. Is the manuscript presented in an intelligible fashion and written in standard English?

Reviewer #1: Yes

Reviewer #2: Yes

5. Review Comments to the Author

Reviewer #1: This study allows us to incorporate elements that demonstrate how social determinants of health affect the probability of dying of newborns in Madagascar in different locations on the island.

As reported by the WHO Commission on Social Determinants of Health, health inequality contributes to poor health outcomes for poor and disadvantaged segments of society and is clearly visible in the results to be published.

Reviewer #2: Global Assessment

The article titled “Dynamics of factors associated with neonatal death in Madagascar: a comparative analysis of the 2003, 2008, 2021 DHS” presents a valuable and comprehensive analysis of neonatal mortality trends in Madagascar across nearly two decades. This work is significant for public health as it uses large, representative samples over several years to analyze factors contributing to neonatal deaths, allowing for essential insights into changes over time. The article is well-structured, easy to read, and provides clear explanations, making findings accessible. It will be a critical resource for policymakers aiming to address neonatal health disparities in Madagascar.

Introduction

The introduction effectively contextualizes neonatal mortality in Madagascar. However, I would suggest integrating lines 93 to 106 into the introduction, as they provide important contextual information that would enhance the reader’s understanding of the study's background from the outset. This section would benefit from a more fluid transition into the objectives and hypotheses of the study, aligning the study’s aims with the country’s public health challenges and goals.

Methods

The methodology is comprehensive but would benefit from a few clarifications and additional information:

• Data Collection Personnel and Process: It is not specified who collected the data (e.g., field researchers, healthcare professionals, etc.) and how this process was conducted. Additional information on the training and qualifications of those collecting data would add clarity, as well as details about quality control procedures during data collection.

• Table 1 - High Response Rate: Achieving a 95% response rate is impressive. More detail on how this rate was maintained, particularly in hard-to-reach areas or among specific demographic groups, would be helpful in understanding any potential selection bias or limitations in the data.

• Line 171 - Classification of Prenatal Visits: The categorization of prenatal visits (less than 3, 4 to 7, more than 8) appears somewhat ambiguous, as it is unclear where women who had exactly 8 visits are classified. A clearer breakdown or explanation would be beneficial.

• Health Indicators: The study could be strengthened by including other important maternal and neonatal health indicators, such as rates of prematurity, intrauterine growth restriction, or maternal health conditions like gestational diabetes and HIV. These factors are often associated with neonatal mortality, and their absence may limit the study's scope.

• Software Mention: There is a redundancy in mentioning “Stata 17®” on lines 183 and 250. This could be corrected for clarity.

• As a considerable portion of births may occur outside formal healthcare facilities, it would be relevant to know if data from traditional birth attendants were available or if there was information on births occurring in non-hospital settings. The distinction between neonatal deaths and deaths occurring in-utero prior to labor is often challenging to make. It would be helpful to clarify how these were differentiated in the dataset or if there are any limitations due to potential classification errors.

Results

The results are well-presented and offer an insightful breakdown of various determinants of neonatal death. A few aspects could be further clarified:

• Definitions of Socioeconomic Indicators: Terms like “very poor households” require more precise definitions, as socio-economic indicators can vary greatly depending on geographic and cultural context. Clarifying the criteria used to define these categories would make it easier to interpret the results.

• Decimal Places in Data: In some sections, results are presented with more than one decimal point, which can make them harder to read. Using only one decimal place throughout would enhance readability.

• “Difficulty Accessing Health Center”: This factor is central to understanding neonatal mortality risks but would benefit from more detail. Is the difficulty due to financial constraints, geographical barriers, or limited health infrastructure? Specifying the components of this variable would enhance the study’s clarity.

• Formatting of Results: For instance, on lines 311–312, where the increase is given in both percentage and times, using a consistent format would improve clarity. Choosing either a percentage or fold-change format throughout the document would enhance readability.

Discussion

The discussion is generally robust, providing valuable insights into how the results can inform health policy. However, a few additions could strengthen this section:

• Territorial Differences: The analysis of neonatal death prevalence by region and time period is very interesting, particularly the significant variability observed in urban settings between 2003 and 2008. However, the discussion does not address potential reasons for these shifts. Including possible explanations for regional differences, or highlighting any changes in healthcare access or policies, would provide valuable context.

• The limitations regarding data quality could be reinforced. Given the nature of the data, quality is crucial and may influence the findings, potentially explaining the observed territorial differences. Additionally, improvements in data quality over time could impact the trends observed in the study. Furthermore, births occurring outside healthcare facilities, especially those assisted by traditional birth attendants, may be missed, leading to possible underreporting in certain areas. Moreover, neonatal deaths occurring shortly after birth are often undocumented or misclassified, which can result in inaccuracies and limit the precision of mortality estimates in the study.

Overall, this article is a valuable contribution to understanding the factors influencing neonatal mortality in Madagascar. Minor clarifications in definitions, methodological details, and data quality considerations would enhance its impact and clarity.

6. PLOS authors have the option to publish the peer review history of their article (what does this mean? ). If published, this will include your full peer review and any attached files.

**Do you want your identity to be public for this peer review?** For information about this choice, including consent withdrawal, please see our Privacy Policy .

Reviewer #1: No

Reviewer #2: **Yes: ** Lison Rambliere

---

## [Decision Letter · Decision Letter 1]

31 Mar 2025

“Dynamics of factors associated with neonatal death in Madagascar: a comparative analysis of the 2003, 2008, 2021 DHS”

PGPH-D-24-02012R1

Dear Dr. Rakotondrasoa,

We are pleased to inform you that your manuscript '“Dynamics of factors associated with neonatal death in Madagascar: a comparative analysis of the 2003, 2008, 2021 DHS”' has been provisionally accepted for publication in PLOS Global Public Health.

Best regards,

Hassan Haghparast Bidgoli

Academic Editor

Thanks for addressing the reviewers's comments. Apart from a minor inquiry from the second reviewer (below), there is no further feedback from the reviewers and the editor.

Reviewer Comments (if any, and for reference):

Reviewer's Responses to Questions

**Comments to the Author**

1. If the authors have adequately addressed your comments raised in a previous round of review and you feel that this manuscript is now acceptable for publication, you may indicate that here to bypass the “Comments to the Author” section, enter your conflict of interest statement in the “Confidential to Editor” section, and submit your "Accept" recommendation.

Reviewer #2: All comments have been addressed

Reviewer #3: All comments have been addressed

2. Does this manuscript meet PLOS Global Public Health’s publication criteria ? Is the manuscript technically sound, and do the data support the conclusions? The manuscript must describe methodologically and ethically rigorous research with conclusions that are appropriately drawn based on the data presented.

Reviewer #2: Yes

Reviewer #3: Yes

3. Has the statistical analysis been performed appropriately and rigorously?

Reviewer #2: Yes

Reviewer #3: Yes

4. Have the authors made all data underlying the findings in their manuscript fully available (please refer to the Data Availability Statement at the start of the manuscript PDF file)?

Reviewer #2: Yes

Reviewer #3: Yes

5. Is the manuscript presented in an intelligible fashion and written in standard English?

Reviewer #2: Yes

Reviewer #3: Yes

6. Review Comments to the Author

Reviewer #2: Thank you very much for your thorough revisions. The changes you have made significantly improve the clarity and readability of the manuscript. Your responses to the comments are highly detailed and precise, which makes it easy to follow your reasoning and the methodological choices you have made.

I would also like to highlight the richness of your work. The study provides valuable insights and represents a meaningful contribution to current knowledge in this field. The depth of your analysis and the way you address key issues make this a particularly relevant and interesting piece of research.

I only have one minor remark: I noticed that the word "usually" appears twice in line 200. Could you confirm whether this repetition is intentional or if it should be adjusted?

Once again, thank you for your efforts in revising the manuscript. I look forward to seeing the final version.

Best regards,

Reviewer #3: Neonatal mortality remains a major public health challenge, despite cost-effective interventions in recent years. The paper explores neonatal mortality in Madagascar. The statistical analysis is sound and the results are properly discussed.

As far as I can see comparing the revised version with the original, the paper was improved.

7. PLOS authors have the option to publish the peer review history of their article (what does this mean? ). If published, this will include your full peer review and any attached files.

**Do you want your identity to be public for this peer review?** For information about this choice, including consent withdrawal, please see our Privacy Policy .

Reviewer #2: **Yes: ** Lison Ramblière

Reviewer #3: No
